

# IL-6 and TNF-α salivary levels according to the periodontal status in Portuguese pregnant women

Vanessa Machado[1], Maria Fernanda Mesquita[2], Maria Alexandra Bernardo[2], Ester Casal[3], Luís Proença[2] and José João Mendes[1]

[1] Clinical Research Unit, Centro de Investigação Interdisciplinar Egas Moniz, Instituto Universitário Egas Moniz, Monte da Caparica, Portugal

[2] Centro de Investigação Interdisciplinar Egas Moniz, Instituto Universitário Egas Moniz, Monte da Caparica, Portugal

[3] Obstetrics and gynecology, Hospital Garcia de Orta, Almada, Portugal

Corresponding author
Vanessa Machado,
vmachado@egasmoniz.edu.pt

## ABSTRACT

**Background.** Periodontitis is associated with increased concentration of inflammatory markers and saliva has been proposed as a non-invasive diagnostic fluid in oral and systemic diseases. The levels of salivary biomarkers, such as cytokines, could potentially be used to distinguish periodontal healthy individuals from subjects with periodontal disease. The purpose of this study was to characterize the salivary levels of two inflammatory biomarkers associated with periodontitis, interleukin-6 (IL-6) and tumour necrosis factor-alpha (TNF-α), in order to assess whether these cytokines salivary levels could potentially be used to complement periodontitis pregnant women diagnose.

**Methods.** Forty-four pregnant women were distributed into three groups, according to their periodontal status: healthy, mild/moderate periodontitis and severe periodontitis. Unstimulated saliva was collected and analysis of TNF-α and IL-6 salivary levels were performed with Immulite®.

**Results.** Women with periodontitis exhibited significantly higher levels ($p = 0.001$) of salivary IL-6 and TNF-α compared with the healthy group: 25.1 ($\pm$11.2) pg/mL vs. 16.3 ($\pm$5.0) pg/mL and 29.7 ($\pm$17.2) pg/mL vs. 16.2 ($\pm$7.6) pg/mL, approximately 1.5 and 1.8 times more, respectively. Additionally, cytokines were significantly increased ($p < 0.05$) in severe periodontitis compared to periodontal healthy pregnant women.

**Conclusions.** These results revealed that IL-6 and TNF-α salivary biomarkers provide high discriminatory capacity for distinguishing periodontal disease from periodontal health in pregnant women.

## INTRODUCTION

Periodontal diseases continue to be a major public health problem worldwide, but there is evidence that the initiation, progression and severity does not affect all people in the same way (*Dye, 2012*; *Petersen & Ogawa, 2012*; *Baelum & López, 2013*; *Persson, 2017*). Some epidemiological studies have demonstrated that gingival inflammation affects 60 to

75% of pregnant women (*Loe & Silness, 1963*; *Silness & Löe, 1964*; *Tilakaratne et al., 2000*; *Michalowicz et al., 2008*; *Ho & Chou, 2016*), although not all present the same gingival inflammatory pattern and the symptoms can range from mild inflammation to severe hyperplasia, pain and profuse bleeding.

There is evidence that support a causal relationship between inflammation and spontaneous preterm labour (*Romero et al., 2006*), but the exact etiology of pregnancy gingivitis is still unknown. Some studies demonstrate that the increase of female sexual hormones' serum concentration during gestation, may lead to gingivitis gestation (*Brabin, 1985*; *Gürsoy et al., 2010*). Further, host immune and inflammatory responses play a major role in periodontitis (*González-Jaranay et al., 2017*) and some cytokines genes were suggested to influence the development of periodontal disease (*D'Aiuto et al., 2004*; *Ebersole et al., 2016*; *Kinane, Stathopoulou & Papapanou, 2017*). Also, when compared to healthy patients, subjects with gingivitis or periodontitis produce high levels of inflammatory mediators, such as IL-6 and TNF-α (*Zhu et al., 2016*).

Saliva, as a pooled sample, contains specific biomarkers for unique pathological aspects of periodontal disease, such as interleukin-1β (IL-1β), IL-6, IL-8, IL-11 and tumor necrosis factor-alpha (TNF-α) (*Graves, 2008*; *Rathnayake et al., 2017*). Recent studies suggested that cytokine levels in this fluid might be linked with the periodontal status of the patient (*Jaedicke, Preshaw & Taylor, 2016*; *Belstrøm et al., 2017*). Thus, qualitative changes in the levels of these biomarkers could have diagnostic and therapeutic significance. Interleukin 6 (IL-6) is a pro-inflammatory cytokine associated with the severity of periodontitis contributing to bone resorption (*Moreira et al., 2007*; *Graves, 2008*; *Jaedicke, Preshaw & Taylor, 2016*; *Zhu et al., 2016*). TNF-α is a pro-inflammatory cytokine that has an effect in the activation of inflammatory leukocytes, modification of vascular permeability and induction of bone resorption (*Assuma et al., 1998*; *Varghese et al., 2015*) and is a main inducer of IL-6 (*Katz, Nadiv & Beer, 2001*).

In the past, some studies demonstrated a correlation between IL-6 and TNF-α with periodontal status of pregnant women in crevicular fluid and gingival tissue (*Carrillo-De-Albornoz et al., 2012*; *Otenio et al., 2012*; *Wu et al., 2016*). However, there are no investigations concerning the salivary levels of these biomarkers in pregnant women. Therefore, the goal of this pilot investigation was to assess the salivary concentration of IL-6 and TNF-α, according to the periodontal status, in a sample of Portuguese pregnant women.

## MATERIALS AND METHODS

### Ethical considerations

This study was approved by a Portuguese state recognized Ethics Committee, from Garcia de Orta Hospital (Ethical Application Ref: 06/2015) and was carried out in accordance with the Helsinki Declaration of 1975 as revised in 2013. Written informed consent was obtained from all participants prior to appointment. All data were registered on a database specifically created for this purpose, where a coded number was attributed to each participant. This was a cross-sectional study without study-defined medical or dental

interventions. Patients with diagnosed pathological conditions were referred to receive appropriate treatment.

## Patient selection

This pilot study was conducted at the Obstetrics and Gynecology Departments of Garcia de Orta Hospital (Almada, Portugal) over one month period (February 2015). Out of the 408 pregnant women that were being attended at those Departments, 82 (20%) were randomly selected to participate. From those, taking into account the exclusion criteria, 44 (10.8% of total) were enrolled in the study. Exclusion criteria were women with congenital uterine and/or vaginal malformations, fetal malformation, multifetal gestation, chronic diseases (e.g., diabetes, hypertension, epilepsy, cardiac disease, lung disease, renal disease, positive test for human immunodeficiency virus (HIV)), history of systemic antibiotic treatment or dental prophylaxis in the previous six months, and using systemic or topical antimicrobial and/or anti-inflammatory therapy within the previous three months and smoking habit. None of the women had received periodontal therapy before and during pregnancy.

## Questionnaire

All women answered a questionnaire, to obtain information about their sociodemographic status (age, marital status, education level, and occupation) and their personal oral hygiene habits (frequency of tooth brushing and dental floss usage).

Educational level was assessed in two categories: basic/middle (1–12 years) and higher (>12 years). Employment status of each participant was classified as: employed or unemployed. Oral hygiene habits were assessed by information about toothbrush frequency (one time daily, two or more times daily) and dental flossing.

## TNF-$\alpha$ and IL-6 measurement in saliva

Unstimulated saliva samples were collected, by passive drooling, in a Falcon® tube for 2 min, between 9:00 am and 11:00 am. Samples were frozen at the collection day and stored at $-80\ ^\circ$C until further analysis. The quantifications in whole unstimulated saliva were performed according to Immulite® (Siemens, Germany) manufacturer's protocol and assessed by a duplicate of each sample analyte. Both TNF-$\alpha$ and IL-6 were detected in all samples. IL-6 and TNF-$\alpha$ levels were expressed in pg/mL.

## Clinical examination

Each subject, who accepted to participate in the study, was assessed by an experienced and calibrated examiner. Clinical examination was performed using a headlight with the individuals seated on a regular chair in Garcia de Orta Hospital and required, on average, 45 min, without radiographic examination.

Periodontitis was defined as severe (individuals with $\geq 2$ interproximal sites with clinical attachment loss (CAL) $\geq 6$ mm, not on the same tooth and $\geq 1$ interproximal sites with probing depth (PD) $\geq 5$ mm), moderate (individuals with $\geq 2$ interproximal sites with CAL $\geq 4$ mm, not on the same tooth or $\geq 2$ interproximal sites with PD $\geq 5$ mm, not on the same tooth) and mild ($\geq 2$ interproximal sites with CAL $\geq 3$ mm, and $\geq 2$ interproximal sites with PD $\geq 4$ mm or one site with PD $\geq 5$ mm) (*Page & Eke, 2007*).

**Table 1  Socio-demographic characteristics and oral health behaviors of subjects according to their periodontal status.**

|  | Healthy ($n = 15$) | | Mild/Moderate periodontitis ($n = 16$) | | Severe Periodontitis ($n = 13$) | |
|---|---|---|---|---|---|---|
|  | *n* | % | *n* | % | *n* | % |
| Education level |  |  |  |  |  |  |
| Basic/Middle | 8 | 53.3 | 13 | 81.2 | 8 | 61.5 |
| Higher | 7 | 46.7 | 3 | 18.8 | 5 | 38.5 |
| Marital status |  |  |  |  |  |  |
| Married | 11 | 73.3 | 13 | 81.2 | 8 | 61.5 |
| Single | 4 | 26.7 | 3 | 18.8 | 5 | 38.5 |
| Occupation |  |  |  |  |  |  |
| Employed | 9 | 60.0 | 11 | 68.8 | 9 | 69.2 |
| Unemployed | 6 | 40.0 | 5 | 31.2 | 4 | 30.8 |
| Toothbrush frequency |  |  |  |  |  |  |
| One time daily | 1 | 6.7 | 3 | 18.8 | 2 | 13.6 |
| Two or more times daily | 14 | 93.3 | 13 | 81.2 | 11 | 86.4 |
| Dental floss usage |  |  |  |  |  |  |
| Yes | 7 | 46.7 | 4 | 25.0 | 3 | 23.1 |
| No | 8 | 53.3 | 12 | 75.0 | 10 | 76.9 |

## Data analysis

Data analysis was performed using IBM SPSS Statistics version 24.0 for Windows (Armonk, NY: IBM Corp.). Descriptive statistics as frequencies, means and standard deviations were calculated. Population means were estimated by calculating 95% confidence intervals (95% CI). Inferential statistics methodologies ($t$-Student's and ANOVA with Brown-Forsythe correction tests) were used to compare both TNF-$\alpha$ and IL-6 data as a function of the periodontal status. The level of significance was set at 5%.

## RESULTS

The patient's characteristics, according to the periodontal diagnosis, are shown in Tables 1–3. No significant differences were found in socio-demographic and oral health behaviour characteristics among the groups. In total, mean age was 32.4 years ($\pm$5.5) (range, 15–43 years) and the average gestation period was 25.4 weeks (range, 6–41 weeks). The majority were in the second (43.2%) and third trimesters (40.9%) of pregnancy, and only 15.9% were in the first-trimester.

Moreover, the education level showed similar numbers among healthy ones, but the majority of pregnant women with mild/moderate or severe periodontitis had basic/middle (81.2%, 61.5% respectively) education levels. The majority of participants were employed (65.1%) and married (72.7%). Concerning the attitude and behaviour of pregnant women, 86.4% participants ($n = 38$) brushed their teeth twice or more a day and 68.2% were not using interdental brushes and dental floss.

**Table 2** Distribution of cytokines salivary levels (pg/ml), presented as mean (±standard deviation) and 95% CI for mean, for healthy and periodontitis subjects.

| Inflammatory cytokines (pg/mL) | Healthy ($n = 15$) | | Periodontitis ($n = 29$) | | $p^*$ |
|---|---|---|---|---|---|
| | Mean (±SD) | 95% CI | Mean (±SD) | 95% CI | |
| TNF-α | 16.3 (±5.0) | [13.5–19.1] | 25.1 (±11.2) | [20.9–29.4] | 0.001 |
| IL-6 | 16.2 (±7.6) | [12.0–20.5] | 29.7 (±17.2) | [23.2–36.3] | 0.001 |

Notes.
$^*t$-Student test.

**Table 3** Distribution of cytokines salivary levels (pg/ml), presented as mean (±standard deviation) and 95% CI for mean, for healthy, mild/-moderate and severe periodontitis groups.

| Inflammatory cytokines (pg/mL) | Healthy ($n = 15$) | | Mild/Moderate Periodontitis ($n = 16$) | | Severe Periodontitis ($n = 13$) | | $p^*$ |
|---|---|---|---|---|---|---|---|
| | Mean (±SD) | 95% CI | Mean (±SD) | 95% CI | Mean (± SD) | 95% CI | |
| TNF-α | 16.3 (±5.0)[a] | [13.5–19.1] | 24.3 (±12.5)[a,b] | [17.6–30.9] | 26.2 (±9.6)[b] | [20.4–32.0] | 0.020 |
| IL-6 | 16.2 (±7.6)[a] | [12.0–20.5] | 27.1 (±18.4)[a,b] | [17.3–36.9] | 33.0 (±15.8)[b] | [23.4–42.5] | 0.016 |

Notes.
*ANOVA with Brown-Forsythe correction. Different lower case letters indicate significant differences between means in the same row (Games-Howell post-hoc test, $p < 0.05$).

The observed prevalence of periodontitis was 65.9% (95% CI [52.6–79.1%]). Specifically, the prevalence of mild/moderate and severe periodontitis was 36.4% (95% CI [23.0–49.8%]) and 29.5% (95% CI [16.8–42.2%]), respectively.

Descriptive statistics (mean and standard deviation) and 95% CI for means, of IL-6 and TNF-α, for healthy and periodontitis groups, were calculated and are displayed in Table 2. Mean salivary levels of IL-6 and TNF-α were significantly higher ($p = 0.001$) in subjects with periodontitis than in healthy subjects: 25.1 (±11.2) vs. 16.3 (±5.0) pg/mL and 29.7 (±17.2) vs. 16.2 (±7.6) pg/mL, approximately 1.5 times and 1.8 times more, respectively.

Cytokine concentrations were significantly different between the healthy and different periodontal status groups (Table 3). Salivary levels of IL-6 and TNF-α were significantly increased ($p < 0.05$) in severe periodontitis compared to periodontal healthy group, but were not found to be statistically significant among periodontitis groups.

## DISCUSSION

The purpose of this pilot study was to evaluate salivary levels of IL-6 and TNF-α according to periodontal status in pregnant women, in order to assess if the level of these pro-inflammatory cytokines could potentially be used as complementary diagnostic in pregnant women with periodontitis. The main finding was that periodontitis was associated with a significant increase in salivary concentrations of all cytokines investigated when compared with periodontal health pregnant women.

This cross-sectional pilot study assessed the periodontal status of forwarded pregnant women subjects who were attended at Obstetrics and Gynecology Departments of Garcia de Orta Hospital, that is located in the metropolitan area of Lisbon. To the best of our

knowledge, this is the first investigation that associated periodontal status and cytokines levels in a Portuguese women pregnant population.

Although our findings are somewhat limited by the small sample size and for being a cross-sectional study, the selection criteria were very narrow and served to avoid potential influence on cytokines levels, thus increasing the strength of the results. Furthermore, it does not provide temporal relationship between exposure and outcome. Thus, in the future we intend to perform a longitudinal study to clarify the periodontal effect and complications on pregnancy.

Traditionally, periodontal diagnosis criteria includes plaque index, gingival index, clinical attachment levels, probing depths, bleeding on probing, mobility of teeth, furcation involvement and radiographic analysis (*Page et al., 1997*; *Eke et al., 2015*). However, pregnant women are not recommended to do radiographic analysis and the diagnosis with these criteria takes a long time. Additionally, these diagnostic parameters are excellent on determining a past history of periodontal disease, however they do not evaluate the inflammatory pattern of the ongoing disease and it is not possible to detect its onset or progression. Saliva represents from whole mouth with all periodontal sites, thereby giving a general assessment of periodontal disease. Thus, salivary cytokine levels have the potential to reflect current activity, disease severity and possibly predict future disease progression, and make aware of immediate or future treatment needs (*Kaufman & Lamster, 2000*; *Prasad, Tyagi & Aggarwal, 2015*; *Jaedicke, Preshaw & Taylor, 2016*; *Korte & Kinney, 2016*; *Morand et al., 2017*).

In this investigation, the periodontitis group showed higher salivary levels of IL-6 and TNF-$\alpha$ compared with periodontal healthy pregnant women. In detail, IL-6 and TNF-$\alpha$ were only significantly increased in severe periodontitis compared to periodontal healthy groups. Moreover, there were differences between mild/moderate and severe periodontitis. These data are in accordance with previous findings where IL-6 and TNF-$\alpha$ salivary concentrations were significantly elevated in periodontitis patients (*Taba Jr et al., 2005*; *Miller et al., 2006*; *Scannapieco et al., 2007*; *Frodge et al., 2008*; *Giannobile et al., 2009*; *Ebersole et al., 2013*). According to our results, IL-6 and TNF-$\alpha$ salivary levels appears to have potential to distinguish pregnant women with and without periodontal disease.

## CONCLUSIONS

Periodontally compromised pregnant women showed significantly higher IL-6 and TNF-$\alpha$ salivary levels than healthy ones. These salivary biomarkers are likely to provide great clinical benefit when supplemented with other clinical information. More studies are needed with longitudinal methodology and larger samples to provide validated reference values that distinguish periodontal disease from periodontal health, especially in initial and developing phases, in order to predict the appearance and estimate the future disease progress during pregnancy.

### Funding

The authors received no funding for this work.

### Competing Interests

The authors received institutional support from the Egas Moniz–Cooperativa de Ensino Superior (Egas Moniz, CRL) and CiiEM Biochemistry Laboratory (BioquiLab). The authors declare that there are no financial or commercial conflicts of interest.

### Author Contributions

- Vanessa Machado conceived and designed the experiments, performed the experiments, contributed reagents/materials/analysis tools, prepared figures and/or tables, authored or reviewed drafts of the paper, approved the final draft.
- Maria Fernanda Mesquita and Maria Alexandra Bernardo conceived and designed the experiments, contributed reagents/materials/analysis tools, authored or reviewed drafts of the paper, approved the final draft.
- Ester Casal conceived and designed the experiments, performed the experiments, contributed reagents/materials/analysis tools, authored or reviewed drafts of the paper, approved the final draft.
- Luís Proença analyzed the data, prepared figures and/or tables, authored or reviewed drafts of the paper, approved the final draft.
- José João Mendes conceived and designed the experiments, authored or reviewed drafts of the paper, approved the final draft.

### Human Ethics

The following information was supplied relating to ethical approvals (i.e., approving body and any reference numbers):

The Hospital Garcia de Orta granted ethical approval to conduct the study within its facilities. Ethical Application Ref: 06/2015.

### Data Availability

Zenodo: https://zenodo.org/record/1179237#.WoxmcZM-fOQ.

### Supplemental Information

Supplemental information for this article can be found online at http://dx.doi.org/10.7717/peerj.4710#supplemental-information.

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
