# Peer review of "IL-6 and TNF-α salivary levels according to the periodontal status in Portuguese pregnant women"

_PeerJ, doi:10.7717/peerj.4710_

## Round 0.1 · original submission · Major Revisions

Please include more details of the experimental design as suggested by the reviewers and address the concerns they have raised.

Reviewer 1 ·

Basic reporting

English used in the manuscript needs improvement. Long ambiguous sentences are used, which make it confusing to read.

Literature references are sufficient.

Professional article structure utilized.

The results of the study are not sufficient to support the hypotheses.

Experimental design

Research is within the aims and scope of PeerJ.

It is not clearly stated how the research question really fills the gap in the knowledge. The question that the authors are asking is whether changes in salivary il-6 and TNF alpha levels can be utilized for diagnosis of periodontal disease in pregnant women. Diagnosis of periodontal disease requires a periodontal probe and measurement of attachment loss around teeth. What the additional benefits are of a salivary test using inflammatory markers is not clearly stated. In the past several studies have looked at the salivary levels of inflammatory cytokines in patients with periodontal disease, yet to date there are no salivary tests available. This discrepancy needs to be clearly addressed in the manuscript. It is also not clear why the authors chose only these two cytokines.

The level of investigation is not rigorous enough. For example, it is not clear at what point during the pregnancy, the saliva samples were collected. At least two to three specific time point based saliva collection and analysis during the course of pregnancy would be ideal. Because within the healthy group there could be natural changes in the inflammatory markers during the course of pregnancy. Also the authors are analyzing only two cytokines. An array of inflammatory markers are needed for more rigorous investigation.

Methods are not described with detail. For example it is not clear if smokers were excluded from this study. Also the time point of saliva collection is not mentioned and the method of saliva collection is not explained in detail.

Validity of the findings

Although there might be a statistical difference in the levels of the cytokines between the healthy and periodontitis groups I am not sure if there is a clinical significance as the differences in cytokine levels are not even twice the normal levels.

Statistical analysis is done by ANOVA. For clinical studies, it would be ideal to perform ODDS Ratio and Relative Risk analyses for more robust statistical outcomes.

Conclusions stated do not support the original question based on the weak methodology and limited results.

Additional comments

Good study but lacks rigorous investigation. More data collection points needed. More inflammatory cytokine profiling needed. Statistical analysis should include Specificity, Sensitivity, Odds Ratio and Relative Risk. English in the manuscript needs professional editing.

·

Basic reporting

The study entitled “IL-6 and TNF-α salivary levels according to the periodontal status in Portuguese pregnant women” is well designed, and has a robust hypothesis introducing the pathogenesis of the association between pregnancy and periodontal disease.

It has been well known that interleukin 1-beta is also an important pro-inflammatory cytokine playing an important role in pregnancy, pregnancy outcomes or pathologies of pregnancy, and periodontal disease. Thus, the authors should also introduce that the role of IL-1 beta in periodontal disease and pregnancy. Just as, IL-1 beta is a precursor and cytokine for IL-6 and TNF-α.

Experimental design

It is a well designed, and the relationship between hypothesis and results is well discussed.
Because of the fact that this is a cross-sectional study, it should be stated as an important limitation of this study.

Validity of the findings

The longitudinally studies suggesting the poor pregnancy outcomes (such as preterm birth, low birth weight, and preterm and low birth weight, etc ) and periodontal disease in the same population will also be helpful to clarify the effect of periodontal disease on pregnancy complications or poor pregnancy outcomes.. These statements should be added to the section of conlusion of the text for further consideration.

Additional comments

1. The study entitled “IL-6 and TNF-α salivary levels according to the periodontal status in Portuguese pregnant women” is well designed, and has a robust hypothesis introducing the pathogenesis of the association between pregnancy and periodontal disease.
2. It has been well known that interleukin 1-beta is also an important pro-inflammatory cytokine playing an important role in pregnancy, pregnancy outcomes or pathologies of pregnancy, and periodontal disease. Thus, the authors should also introduce that the role of IL-1 beta in periodontal disease and pregnancy. Just as, IL-1 beta is a precursor and cytokine for IL-6 and TNF-α.
3. It is a well designed, and the relationship between hypothesis and results is well discussed.
4. Because of the fact that this is a cross-sectional study, it should be stated as an important limitation of this study.
5. The longitudinally studies suggesting the poor pregnancy outcomes (such as preterm birth, low birth weight, and preterm and low birth weight, etc ) and periodontal disease in the same population will also be helpful to clarify the effect of periodontal disease on pregnancy complications or poor pregnancy outcomes.. These statements should be added to the section of conlusion of the text for further consideration.

---

## Round 0.2 · accepted · Accept

I can confirm that all the queries of the reviewers have been sufficiently addressed.